# Emergence of SARS-CoV-2 Variants in the World: How Could This Happen?

**DOI:** 10.3390/life12020194

**Published:** 2022-01-28

**Authors:** Alfredo Parra-Lucares, Paula Segura, Verónica Rojas, Catalina Pumarino, Gustavo Saint-Pierre, Luis Toro

**Affiliations:** 1Division of Critical Care Medicine, Department of Medicine, Hospital Clínico Universidad de Chile, 8380456 Santiago, Chile; alfredop@ug.uchile.cl (A.P.-L.); vrojas@hcuch.cl (V.R.); 2Department of Anatomic Pathology, Hospital Clínico Universidad de Chile, 8380456 Santiago, Chile; psegura@hcuch.cl; 3Centro de Investigación Clínica Avanzada, Hospital Clínico Universidad de Chile, 8380456 Santiago, Chile; 4School of Medicine, Faculty of Medicine, Universidad de Chile, 8380456 Santiago, Chile; catalinapumarino@gmail.com; 5Microbiology Unit, Clinical Laboratory, Hospital Clínico Universidad de Chile, 8380456 Santiago, Chile; gsaintpierre@hcuch.cl; 6Division of Nephrology, Department of Medicine, Hospital Clínico Universidad de Chile, 8380456 Santiago, Chile; 7Critical Care Unit, Clínica Las Condes, 7591047 Santiago, Chile

**Keywords:** SARS-CoV-2, COVID-19, SARS-CoV-2 variants, epidemiology, diagnosis, prognosis, social determinants of health, developing countries

## Abstract

The COVID-19 pandemic has had a significant global impact, with more than 280,000,000 people infected and 5,400,000 deaths. The use of personal protective equipment and the anti-SARS-CoV-2 vaccination campaigns have reduced infection and death rates worldwide. However, a recent increase in infection rates has been observed associated with the appearance of SARS-CoV-2 variants, including the more recently described lineage B.1.617.2 (Delta variant) and lineage B.1.1.529/BA.1 (Omicron variant). These new variants put the effectiveness of international vaccination at risk, with the appearance of new outbreaks of COVID-19 throughout the world. This emergence of new variants has been due to multiple predisposing factors, including molecular characteristics of the virus, geographic and environmental conditions, and the impact of social determinants of health that favor the genetic diversification of SARS-CoV-2. We present a literature review on the most recent information available on the emergence of new variants of SARS-CoV-2 in the world. We analyzed the biological, geographical, and sociocultural factors that favor the development of these variants. Finally, we evaluate the surveillance strategies for the early detection of new variants and prevent their distribution outside these regions.

## 1. Introduction

The pandemic due to the coronavirus disease 19 (COVID-19) has had a significant impact worldwide, with more than 280,000,000 infected worldwide and more than 5,400,000 deaths by 31 December 2021 [1]. This disease is caused by the severe acute respiratory syndrome coronavirus 2 (SARS-CoV-2), a virus of the *Coronaviridae* family, which infects animals and humans [2,3]. Since December 2020, variants of the original SARS-CoV-2 have been identified, which have a significant public health impact, with changes the disease transmissibility and have potential risk to decrease the efficacy of the methods of prevention of contagion especially the efficacy of vaccines [4,5,6]. Although these variants have been described in multiple regions, many of them have developed in developing countries. Recently highlighting the appearance of the lineage B.1.617.2 (Delta variant) and lineage B.1.1.529/BA.1 (Omicron variant) [7,8,9,10].

This review aims to explore the epidemiological and virological characteristics of the SARS-CoV-2 variants with the most significant epidemiological impact. It includes the mechanisms involved in the changes in these variants’ genome and phenotypes. In addition, we will explore the different predisposing factors for these variants to arise in less developed areas. Finally, we will discuss the measures that national and international organizations develop to prevent the emergence and spread of these variants.

## 2. Classification of SARS-CoV-2 Variants

The World Health Organization has established several definitions to evaluate the SARS-CoV-2 variants regarding their changes concerning the original native form [11,12]. Table 1 shows the details of the current definitions.

To classify SARS-CoV-2 variants, different nomenclatures are used. For example, the GISAID, Nextstrain, and Pango classifications refer to genetic lineages. In addition, the use of letters from the Greek alphabet has been recommended for practical use, especially in non-scientific groups [11,13] (example: lineage B.1.617.2—Delta variant).

The main classifications to evaluate the virological and clinical relevance of the different variants are Variants of Concern (VOC) and Variants of Interest (VOI). These classifications are associated to the performance of certain behaviors by the WHO and the country where they were detected (this will be discussed later). Other classifications use by the WHO are Variants Under Monitoring (VUM) and Previously Monitored Variants (Table 1).

As of 31 December 2021, 2 SARS-CoV-2 variants are included as VOIs: lineage C.37 (Lambda variant) and lineage B.1.621 (Mu variant). In addition, 5 SARS-CoV-2 variants are included as VOCs: lineage B.1.1.7 (Alpha variant), lineage B.1.351 (Beta variant), lineage P.1 (Gamma variant), lineage B.1.617.2 (Delta variant), and lineage B.1.1.529/BA.1 (Omicron variant). The general characteristics of these VOCs are presented in Table 2.

It should be noted that SARS-CoV-2 variants can change their classification concerning virological and epidemiological discoveries. For example, the lineages B.1.427/B.1.429 (Epsilon variant), P.2 (Zeta variant), and P.3 (Theta variant) were initially considered as VOIs [11,12]. However, laboratory studies and epidemiological behavior caused it to be reclassified as previously monitored variants. In addition, the lineage B.1.617.2 (Delta variant) was initially considered VOI before switching to VOC [11].

## 3. Virological Characteristics of SARS-CoV-2

SARS-CoV-2 is a virus of the *Coronaviridae* family of the betacoronavirus B lineage [14,15,16]. Coronaviruses are viruses with a single-stranded positive-sense RNA (ssRNA) and a nucleocapsid with helical symmetry. Viruses of the *Coronaviridae* family have been identified in multiple species, both in mammals and birds [17]. To date, 7 species of coronaviruses capable of infecting humans have been identified. 4 of them are associated with upper respiratory tract infection (HCoV-229E, HCoV-NL63, HCoV-OC43, and HCoV-HKU1). These viruses generate upper respiratory manifestations such as the common cold or viral rhinosinusitis, self-limited and exceptionally severe in humans, and contribute to 15–30% of cases of common colds in adults [18]. The other 3 coronaviruses are those that cause lower respiratory infection. These include the severe acute respiratory syndrome-related coronavirus (SARS-CoV or SARS-CoV-1), associated with civets and bats as intermediate animals; the middle east respiratory syndrome-related coronavirus (MERS-CoV), associated with camelids as an intermediate animal; and the recently discovered SARS-CoV-2, of which the intermediary between bats and human infection is not yet known [19,20]. These viruses can produce a systemic inflammatory response syndrome with respiratory distress and the possibility of serious complications, such as respiratory failure and death [20].

The RNA sequence of SARS-CoV-2 has approximately 30,000 nucleotides [21]. This sequence codes mainly for proteins required for the synthesis and structure of the virus [14,22]. The genome of the virus is covered by a nucleocapsid protein (N) [23], which makes up the viral capsid made of 3 structural proteins: membrane protein (M) [24], an envelope protein I [25,26], and spike (S) protein [27,28,29]. Figure 1 summarizes the main characteristics of the structural proteins. In addition to these structural proteins, the virus contains 16 non-structural proteins (nsp1-16), which participate in different functions within the viral replication process [14,22].

The spike protein is a 1273 amino acid transmembrane glycoprotein. It has an exposed N-terminal ectodomain, a transmembrane helix, and a short intracellular C-terminal tail [27]. This protein forms a homotrimer that protrudes through the viral capsid. It comprises 2 functional subunits, regions S1 and S2. The S1 subunit is composed by the N-terminal domain (NTD) and the C-terminal domain (CTD), which constitute the receptor-binding domain (RBD). This structure binds to the host cell receptor, angiotensin-converting enzyme 2 (ACE2), allowing the virus to enter the cell. In addition, RBD includes a receptor-binding motif (RBM), which contains amino acid residues that directly contact ACE2 [30,31]. The S2 subunit includes the fusion peptide (FP), 2 heptad repeat (HR1 and HR2) subdomains, a transmembrane helix, and a cytoplasmic tail. This structure allows the fusion of the virus membrane with the host cell, where FP enters the host cell membrane and destabilizes it, allowing the virus to enter the cell [14,32].

Among these proteins, the spike protein is one of the most relevant molecules in studying SARS-CoV-2 variants. This is due both to its role in viral tropism and the entry of the virus into the host cell and because it is the main target of anti-SARS-CoV-2 vaccines, especially nucleic acid vaccines [28,29]. Therefore, mutations involving this protein can potentially modify the virus infectivity, the severity of the clinical disease, and the efficacy of anti-SARS-CoV-2 vaccination.

## 4. Relevant Characteristics of the SARS-CoV-2 Genome

According to the GISAID (Global Initiative on Sharing Avian Influenza Data) registry, as of 31 December 2021, more than 6,500,000 genomic sequences of the virus have been obtained from different regions of the planet [33]. Homology studies show that, when comparing SARS-CoV-2 with other coronavirus species, it has the highest homology with BatCoV RaTG13, a coronavirus from the bat *Rhinolophus affinis* (96% sequence homology) [34]. This homology supports the hypothesis of an origin of SARS-CoV-2 derived from this coronavirus. However, when comparing SARS-CoV-2 with coronaviruses that infect humans, the highest homology is with SARS-CoV-1, with a sequence homology close to 79% [34,35].

SARS-CoV-2 has a sequence of approximately 29.8 kb of linear single-stranded positive-sense RNA. The sequence presents 12 open reading frames (ORFs), which code for the 27 proteins of the virus. It has an estimated GC content between 32–43%. The genomic organization is 5′-leader sequence-ORF1/ab-S-ORF3a-E-M-ORF6a-ORF7a-ORF7b-ORF8-N-ORF10-3′ [34,36].

Regarding the genetic diversity of SARS-CoV-2, a study evaluating hospitalized patients in China in early 2020 found a mutation rate of 1.1–6.2 × 10^−3^/site/year [37]. More recent studies suggest a lower mutation rate, between 1.5–1.7 × 10^−3^/site/year [38]. This rate is similar than other RNA viruses such as seasonal influenza (H1N1, H3N2) with a rate between 0.6–2.0 × 10^−3^/site/year [37]. Coronaviruses encode a unique proofreading activity that is not found in other RNA viruses, this activity is encoded in the N-terminal exoribonuclease domain of nsp14, which together with nsp10 forms an RNA proofreading complex [39]. Furthermore, a recent study indicated that the mutational dynamics of SARS-CoV-2 vary in different regions. It was observed that in areas where there was a lower control of the replication rate of the virus, there was a higher frequency of mutations than those with better control of expansion [40]. This suggests that the effect of environmental measures and non-pharmacological strategies would make it possible to reduce virus mutations, thus preventing the emergence of new VOIs/VOCs (this will be discussed later).

Regarding the regions of the genetic sequence with the highest mutation rate, a meta-analysis that combined more than 62 studies with 368,316 sequenced genomes showed more significant mutations for the S segment, followed by the N segment [41]. The main mechanisms of the genetic diversity of the virus are random mutations and genetic recombination [34,42,43]. Among these alterations in the genetic sequence, the most relevant are those of the S sequence, which codes for the spike protein. This is because this protein is responsible for the binding of the virus to the target cell and is also the main target of mRNA-based vaccines and adenovirus vectors [28,29]. Therefore, mutations in the S sequence are especially relevant to assess the potential of a new SARS-CoV-2 variant as a probable VOC.

## 5. Genetic Changes in the SARS-CoV-2 Variants of Interest (VOIs) and Variants of Concern (VOCs)

The main characteristic of the SARS-CoV-2 variants identified as VOIs/VOCs is the presence of mutations that can potentially change the virulence of the virus, increase clinical severity, or decrease detection in diagnostic tests [11,12]. Therefore, the primary mutations of interest involve the spike (S) protein, although mutations have also been described for the M, E, N, and ORF sequences. Figure 2 show the list of characteristic mutations (present in more than 75% of the lineage sequences) detected in the 5 VOCs identified by the WHO as of 31 December 2021. Next, we present some relevant data of these 5 VOCs.

### 5.1. Lineage B.1.1.7 (Alpha Variant)

In November 2020, a new variant of SARS-CoV-2 was identified in the United Kingdom based on a sample obtained in September 2020. This new variant stood out from those previously observed due to the more significant increase in cases and the high number of observed mutations of potential epidemiological impact [44]. On 18 December 2020, it was designated as a VOC, and later it was called the Alpha variant, mainly for communicational purposes [11]. This variant has at least 22 characteristic mutations, including 13 non-synonymous mutations, 6 synonymous mutations, and 3 deletions [45]. These mutations include the N501Y mutation of the spike protein, which increases the affinity of RBD with the host cell’s receptor (ACE2), the P681H mutation, which would facilitate the entry of the virus into the cell [44], and the D614G mutation in the spike protein which increases infectivity [46]. The D614G mutation was initially detected during early 2020 and has also been identified in all the VOCs that have appeared after lineage B.1.1.7 [47].

Concerning transmissibility, this variant has a higher transmission power than previous variants of the virus, initially reported between 50–100% higher [48,49]. However, subsequent analyses indicated that the transmissibility increase was lower as previously described [50]. This difference could be associated to the impact of other variables such as health determinants and the scarce availability of supplies such as masks (see below). Regarding the severity of the infection, the evidence is not conclusive. Although initial studies have associated this variant with a higher rate of severe COVID-19 and death [51,52], other studies have shown no differences in adverse outcomes [53]. Finally, regarding the efficacy of vaccines, the available evidence indicates that there would be no decrease in efficacy to prevent infection and adverse outcomes [54,55].

### 5.2. Lineage B.1.351 (Beta Variant)

In December 2020, this new variant was detected in South Africa, identified for the first time in September 2020 in the same country, which presented a rapid expansion in the region as compared to previous variants of the virus [56,57], receiving the VOC designation on 18 December 2020 [11]. This variant has at least 18 characteristic mutations, including some previously described in the Alpha variant, such as N501Y in protein S [58]. Among the mutations of this variant, K417N and E484K stand out, present in the RBM region of the RBD domain of the spike protein, which are associated with increased infectivity [59].

This variant presents higher transmissibility compared to other previous variants [57,59]. Regarding the severity of the infection, as with variant B.1.1.7, the evidence is not conclusive. Epidemiological studies have shown that it would be associated with a more significant number of hospitalizations. However, an increase in mortality has not been observed [60,61]. Finally, on the efficacy of vaccines, the available data indicate that they prevent infection and adverse outcomes [54,55], but some reports indicate less efficacy to prevent infection as compared to previous variants [62].

### 5.3. Lineage P.1 (Gamma Variant)

At the beginning of January 2021, this variant was detected for the first time in Tokyo, Japan, in 4 people from the Brazilian Amazon in December 2020. Its extension was subsequently demonstrated in Brazil [63]. On 11 January 2021, it was designated as VOC by the WHO [11]. This variant has at least 23 characteristic mutations, including some reported in other variants such as N501Y and E484K in protein S, which increase the infective capacity of the virus on the host cell [59,64].

Regarding transmissibility, data reported from Brazil indicate higher transmissibility as compared to other previous variants [63,65]. However, it has not been demonstrated an increase in the severity of the disease. Although an increase in deaths was initially reported in Brazil [66], epidemiological data from Europe do not show an increase in deaths but an increase in hospitalizations [60]. Therefore, it has been proposed that the higher mortality in Brazil was more associated with the region’s socioeconomic and public health limitations [67,68]. Finally, on the efficacy of the vaccines, there is less information as compared to B1.1.7 and B.1.351 lineages. However, the available data indicate that the vaccines prevent infection and adverse outcomes [54,55], similar to variant B.1.351 [56].

### 5.4. Lineage B.1.617.2 (Delta Variant)

On 24 March 2021, the Indian Ministry of Health reported a new SARS-CoV-2 variant from December 2020 samples, with mutations associated to potential immune escape [69]. On 4 April 2021, the WHO assigned the VOI classification, and on 11 May 2021, its classification was changed to VOC [11]. This variant presents at least 21 characteristic mutations, such as D614G, L452R (which gives higher stability to the RBD-ACE2 complex, increasing infectivity), and P681R (which optimizes spike protein cleavage, with the potential to increase transmissibility) [70,71,72]. All these mutations provide a higher infective capacity of the virus than the previously reported variants.

The transmissibility of lineage B.1.617.2 is higher than reported in previous variants, including VOCs previously described [56,73], with high transmissibility indoors [74]. Therefore, as of 31 December 2021, it is the dominant variant of SARS-CoV-2 worldwide [75]. Furthermore, regarding severity, it has been observed that patients infected with this variant have a higher rate of adverse outcomes, including hospitalizations, ICU requirements, and deaths, compared to those infected with variant B.1.1.7 [76,77,78]. Finally, vaccines have a slight decrease in their efficacy to prevent infection but similar efficacy to prevent severe disease when compared with other variants, including previous VOCs [55,56]. For example, a clinical study showed that the BNT162b2 vaccine (Pfizer-BioNTech) had an efficacy of 88% for the delta variant versus 93% for the alpha variant. In comparison, the ChAdOx1 nCoV-19 vaccine (Oxford-AstraZeneca) had an efficacy of 67% for delta variant versus 74% for alpha variant [79].

### 5.5. Lineage B.1.1.529/BA.1 (Omicron Variant)

On 11 November 2021, a new variant of SARS-CoV-2 was reported in South Africa which was detected on a traveler from Botswana. Later, this variant began to be detected in patients from South Africa [9]. The appearance of this variant was associated with an abrupt increase in COVID-19 cases in the country, from an average of 280 daily cases to 800 daily cases in the following weeks. Especially relevant was the situation in the Gauteng province of South Africa, where the doubling time of cases decreased as compared to previous COVID-19 waves in the area (1.2 days versus 1.3–1.7 days) [9,80]. On November 24, 2021, this variant was classified as VUM, and on November 26, the WHO reclassified this new variant as VOC [11].

This variant presents a higher number of mutations as compared to other VOCs, having been identified at least 48 characteristic mutations by 31 December 2021, including several mutations previously not described in other variants [81,82]. Among the most relevant mutations are RBD mutations, which would confer increased binding to ACE2 and infectivity of the host cell [82,83,84].

Due to the short time that has elapsed since this new variant was identified, there is little epidemiological evidence on transmissibility, the severity of infection, and response to vaccines. Therefore, the available information is based mainly on preliminary data and non-peer-review publications, which may change in the short and medium term.

Epidemiological data suggest that lineage B.1.1.529 could infect between 3 and 6 times more people than lineage B.1.617.2 [80,85], making it the variant of SARS-CoV-2 with the highest transmissibility identified up to the date. Given this, lineage B.1.1.529 can become the dominant variant worldwide in short to medium term, surpassing lineage B.1.617.2. Regarding severity, preliminary data suggest that it would be lower than previous variants [85,86]. Data from the United Kingdom suggest that hospitalization rates would be 50–70% lower than for variant B.1.617.2 [86]. There is still little information on rates of severe COVID-19 and deaths. However, its higher transmissibility suggests that the number of hospitalizations would remain high, with the potential risk of a collapse of health systems. Finally, regarding the efficacy of vaccines, preliminary data suggest that vaccines would continue to be effective in preventing infection and adverse outcomes [87]. Reports from Pfizer-BioNTech [88] and AstraZeneca [89] indicate that a vaccine booster produces neutralizing antibodies to the B.1.1.529 variant. However, recent data show that the reinfection rate of lineage B.1.1.529 in COVID-19 survivors would be higher than lineage B.1.351 or B.1.617.2 [90]. This suggests that the immune response after infection or vaccination could be lower than other variants. To evaluate this, more clinical studies are needed.

In December 2021, the Pango Network recommended dividing the original classification of lineage B.1.1.529 into 2 sublineages, BA.1 (the original Omicron variant) and BA.2 [91]. This change would be due to the differences in the nucleotide sequences, where the BA.2 sublineage would not have the del69-70, which could affect the detection by PCR of the virus [91]. The epidemiological impact of BA.2 sublineage is still unknown. In the last week of December 2021, an increase in total COVID-19 cases has been reported, reaching more than 1,400,000 daily worldwide [92,93].

It should be noticed that for 31 December 2021, the two most prevalent SARS-CoV-2 variants are Delta and Omicron, while variants Alpha, Beta, and Gamma have significantly reduced, being practically absent in current COVID-19 patients evaluated [94].

## 6. Predisposing Geographical, Environmental, and Genetic Conditions for the Development of SARS-CoV-2 Variants

SARS-CoV-2 is an RNA-type virus, and therefore, one of the main characteristics given its structure corresponds to the high capacity to mutate because of the low or no correction activity of the viral polymerase protein responsible for the nucleotide synthesis [95,96]. Furthermore, within the predisposing variables to increase its mutation, and, therefore, the emergence of new variants, it has been reported that geographic factors could predispose to these events.

Territorially speaking, the geographical separation between communities tends to generate variants of the virus. This has been reported for other viruses such as influenza, since seasonally, one of the hemispheres develops a variant, and later, it is transmitted to the other hemisphere [97,98]. This is also due to the RNA structure of the influenza virus, with a high mutation capacity, which is why vaccination campaigns with vaccines including new influenza variants are performed every year [99]. This has already been observed in some studies that showed these differences between the different continents. The evaluation of these geographical differences has been detected due to initiatives such as GISAID, a database that initially grouped many sequenced influenza genomes. During the COVID-19 pandemic, this database has also served as a platform to store information regarding SARS-CoV-2 at a global level. This database has allowed us to compare the variation of the viral genome in various regions of the world [100]. These data have revealed that even though the different variants are spread over the six continents, the frequency of several mutations vary in different regions [101]. Therefore, geographic barriers can determine the appearance of new strains within the same continent, given the characteristics of geographic population distribution. This has happened in India with lineage B.1.617.2 (Delta variant) or in South Africa, as is currently happening with lineage B.1.1.529/BA.1 (Omicron variant). In these regions, cases were initially reported in these countries with reduced access to vaccines, especially in more remote populations, and later they became the main variants in their respective territories, rapidly surpassing the dominant variants up to that time [102,103]. In addition, within the same country, clusters of different variants can be generated due to geographical barriers. This has been observed in Chile, a long and narrow country in South America with multiple geographic landmarks, including multiple zones over 4000 m high, multiple islands, deserts, and glacial regions in its more than 4000 km length [104].

Additional to geographical factors, certain biological factors can also explain the appearance of new variants of SARS-CoV-2. These factors can be categorized in two types: factors intrinsic to the virus, which refers to its mutagenic property, and which produces modifications in its structure at the time of replication, giving it comparative advantages to its predecessor versions. For example, in cell-binding sites, such as the spike protein, which, when modified given the mutations of this RNA virus, allows optimizing its affinity to the angiotensin-converting enzyme 2 (ACE2) receptor, generating higher transmissibility and infective capacity [105,106]. This strategy is used by other RNA viruses such as influenza and HIV. However, in the case of SARS-CoV-2, the speed with which mutations occur stands out, mainly due to the corrective capacity it has in its enzymatic system [37,107].

In the same way, we can identify biological factors external to the virus, such as the selective pressure that the immune system exerts on viral behavior and immune escape. Therefore, the virus develops modifications that increase the infectivity of host cells or hide from the immune response, generating mechanisms to escape the neutralizing antibodies developed after vaccination or after COVID-19. This has been demonstrated in several reports of the cross-reactive ability of antibodies against previous and recent variants such as B.1.617.2. The latter shows an increased evasion capacity of the humoral immune system [108,109,110,111,112].

In addition, we can mention the environmental factors that could contribute to the appearance of SARS-CoV-2 variants. Some studies during the appearance of the SARS-CoV-1 virus in China in 2003 already showed that environmental conditions could influence the virus’s transmissibility, infectivity, and stability [113]. Likewise, it is described that there would be favorable climatic conditions for the increase in transmissibility (for example, vectors carried on the wind) and, therefore, potentiates the generation of mutations in different viruses, including SARS-CoV-2. For example, the first large wave of cases in the Northern Hemisphere countries such as China, the US, and Europe during winter and early spring 2020, with a rapid viral expansion and identifying different variants as the months passed. Low temperatures decreased air humidity, and increased air pollution with particulate matter from large cities probably contributed to this process, as well as the survival probabilities of the virus in more temperate climates [114,115,116]. Finally, animal models have been used to evaluate the effect that environmental conditions generate on various tissues, especially the respiratory tract. For example, exposure to particulate matter increases the expression of ACE2 in affected cells, which would enhance interaction of the virus with the host cell [117].

## 7. Influence of Social Determinants on the Development of Variants of SARS-CoV-2

The World Health Organization (WHO) defines social determinants of health as the circumstances in which people are born, grow, work, live and age, including the broader set of forces and systems that influence the conditions of life every day [118,119]. These forces and systems include economic policies and systems, development programs, social norms and policies, and political systems. These conditions can be highly different for various subgroups of a population and can lead to differences in health outcomes. Therefore, they are considered inequalities, just as these differences may be unnecessary and avoidable, and appropriate goals for policies designed to increase equity [118,119,120].

### 7.1. Social Inequality

Regardless of the transmissibility or severity of lineage B.1.1.529/BA.1 (Omicron variant), its appearance almost two years after the start of the COVID-19 pandemic is a demonstration that it is not yet fully controlled. For this reason, attention should be paid to the WHO motto that “none of us is safe until we are all safe” [121]. In this regard, the COVID-19 epidemic has revealed the effect of social inequity. Therefore, these factors influence the appearance of new SARS-CoV-2 VOIs and VOCs. This fact is evidenced by the rapid spread of the lineage B.1.1.529/BA.1, especially among younger patients in South Africa, which has once again put global health systems on alert [80,121]. Inequity has numerous effects on public health, many of which correspond to factors that contribute both to the spread of the COVID-19 pandemic and the development of new variants. For example, in the United States, in Chicago, it has been reported that inequity, social vulnerability, and socioeconomic risk factors in African American groups negatively associated with an increased COVID-19 mortality [122].

### 7.2. Poor Access to Health Systems—Vaccination Effects

Inequality is a factor that influences access to health systems. In the case of the COVID-19 pandemic, this is reflected in the vaccination rates. While in Europe, an average of 60% of the population has complete immunization for COVID-19, reaching 80% only in the United Kingdom, in Africa only 5–10% of the population has received the first dose, of which 24% are found in South Africa [121,123].

Although initiatives such as the COVID-19 Vaccine Global Access initiative (COVAX) have improved access to vaccination in all countries, there is still a significant number of vulnerable populations without vaccination [124,125]. Furthermore, the larger the amount of SARS-Cov-2 viral load, the higher the probability that the virus will mutate and the larger the risk of development of new VOCs and potential vaccine-resistant variants. In conclusion, the emergence of the lineage B.1.1.529/BA.1 and its rapid spread reflects the uneven distribution of COVID-19 vaccines globally, contributing to prolong the pandemic. For this reason, unless equitable access to vaccines is guaranteed, new variants may continue to emerge and spread worldwide [123,124,125]. Given the continued circulation of SARS-CoV-2 and the emergence of VOCs in Africa and elsewhere, vaccine development must adapt and evolve, which involves synergistic work by both vaccine manufacturers and governments, which must guarantee their access [121].

### 7.3. Health Education

Vaccination is undoubtedly the strongest strategy to address this pandemic, and early data suggest that a third booster dose prevents Omicron symptoms in 57% of people [88,89,126]. However, another fundamental aspect of access to health systems is education, which constitutes a significant factor in health promotion and prevention. In addition, hand hygiene, the mandatory use of masks and social distance are still relevant in family and work settings. Regarding the use of masks, both the inequity in access and the asymmetric policies of governments due to their universal use have generated an increase in cases and disproportionate spread in some regions of the world [127]. Therefore, risk communication must be clear, precise, and reliable, guaranteeing the ability to implement evidence-based measures to control the spread of the virus [123].

A crucial aspect of health education is connectivity, which represents an underestimated element of the public health response. The rural population and the elderly are particularly vulnerable since, in addition to the geographical and physical factors that hinder access to health centers, they face the difficulty of the digital knowledge that prevents their access to prevention measures [128,129].

### 7.4. Social Capital

In sociology, social capital is defined as the socio-structural resources that constitute a capital asset for the individual and facilitate certain actions of individuals who are within that structure [130]. This determinant provides numerous benefits during crisis scenarios, where communities with high social capital respond more effectively than those with low social capital. For this reason, some authors argue that the response and recovery from the COVID-19 pandemic may be hampered in many American communities by deficiencies or disruptions in social capital caused by physical distancing [131].

Components of social capital include virtual communities, fostering solidarity, and building trust networks among decision-makers, health workers, and general population [132]. It is plausible to consider that all these elements, which are essential for providing health services, can contribute, at least in part, to preventing the appearance of new VOCs.

### 7.5. Overcrowding

Among the factors that increase exposure to SARS-CoV-2 and, therefore, the development of new variants within economically disadvantaged people is the increased probability of living in overcrowded housing. For example, 7% of the poorest 20% of UK households live in overcrowded housing, which, associated with poor housing conditions, poor access to outer space, and overcrowding will reduce compliance with social distancing [133].

### 7.6. Labor and Economic Conditions

Confinement is quite effective to contain the virus if the entire population respects it despite the high social and economic costs [134]. Therefore, it is complicated for the low-income people, who are often employed in occupations in retail and service sectors, with few options for telecommuting and more likely to have precarious working conditions and unstable incomes. Such financial uncertainty disproportionately damages the mental health of those belonging to low socioeconomic groups, increases stress by weakening the immune system and increasing the likelihood of risky behaviors for health, increasing viral circulation and the risk of development of new VOCs [133].

## 8. Strategies to Control Potential Variants of Concern of SARS-CoV-2

The WHO has established specific actions against a new SARS-CoV-2 variant identified as a potential or confirmed variant of interest (VOI) or variant of concern (VOC) [12]. Table 3 details the main public health activities to be carried out in the face of the identification of VOIs/VOCs. These activities can be summarized into primary activities for WHO and the country that identified the new variant. In relation to the WHO, after the identification of a probable VOI/VOC, it must carry out a comparative analysis of the characteristics of the variant and public health risks, with the collaboration of the other member states, and maintain a global follow-up of this variant. Concerning the country that reports the new VOI/VOC, it must send the available information to a public access database, report the cases and outbreaks associated with the expansion of the VOI/VOC, and perform research to evaluate the impact of the VOI/VOC at the epidemiological level (pathogenicity, severity, response to treatments, among others) [12,135].

### 8.1. Genomic Surveillance of SARS-CoV-2

The SARS-CoV-2 genomic surveillance has been the largest genomic sequencing project ever conducted for a pathogen, with more than 2.8 million sequences completed by August 2021 [136]. This surveillance has had a significant role in the prevention and control of SARS-CoV-2 since it has allowed: a follow-up of the place and moment of origin of the introductory events of mutations and VOIs/VOCs, the characterization of the dissemination of VOIs/VOCs at a regional and international level, and identification of risk factors for the transmission, infectivity, and pathogenicity of the virus. In addition, the search for specific mutations has made it possible to evaluate all virus variants in the spike proteins and in the RBD segment to determine potential mutations with a higher risk of transmissibility and pathogenicity. For example, this knowledge allowed the rapid identification of the lineage B.1.1.529 as a VOC in November 2021 [9,80]. Furthermore, it has been essential for the rapid design of anti-SARS-CoV-2 vaccines and the in vitro evaluation of the efficacy of different vaccines to prevent infection and adverse outcomes associated with COVID-19 [6,87].

However, there are limitations to this system. Given the public health determinants mentioned above, of the total identified SARS-CoV-2 genomes, more than 50% come from the United States and England, contributing less than 10% from developing countries, regions where the most recent VOCs of the virus have formed [14]. This reduces the detection of these new variants, limiting their early detection and the possibility of controlling the spread of the virus. Due to this, it has been insisted on closer international collaboration, the democratization of vaccine access (covering regions with less availability of vaccines) and sharing relevant information more quickly among the different members of the health teams, both public and private. This monitoring has allowed the implementation of public health measures that previously would have been impossible in the face of different infectious outbreaks and may have prevented the infection and deaths associated with COVID-19.

### 8.2. International Anti-SARS-CoV-2 Vaccination Campaign

Effective vaccination is the most effective measure to prevent SARS-CoV-2 infection and viral load at the population level. As of 31 December 2021, 10 vaccines have been approved by WHO for use in patients, with efficacy demonstrated in randomized clinical studies [29,54,55,79,137]. In addition, more than 9,000,000,000 doses of vaccines have been used worldwide [1]. The use of vaccines, especially completing the full vaccination (2 doses for most available vaccines), has been associated with a decrease in cases but specially to a decrease in severe COVID-19, including hospitalizations and deaths. Assuming a homogeneous distribution of vaccinations, this number of vaccinations would indicate that in an estimated world population of 7,900,000,000 by the end of 2021 [138], the entire world population would have at least 1 dose of vaccine. However, the distribution of vaccines has been highly heterogeneous [137,139]. While Western European countries, Japan, Canada have complete vaccination rates above 70%, in regions such as India, South Africa, and Latin America, the complete vaccination rate is below 50%. For example, as of 31 December 2021, India has a 42% complete vaccination rate, Botswana has a 43% rate, Brazil has a 67% rate, and South Africa has a 26% rate. This is associated with the socioeconomic status of the country. While North America, Western Europe, China, and Japan have the highest complete vaccination rates, Africa has the lowest rates globally, with more than 15 countries with vaccination rates below 10% [139].

This vaccination mismatch predisposes to the development of new SARS-CoV-2 VOIs/VOCs. Therefore, the WHO has insisted on global vaccination as one of the primary methods for preventing the development of new variants of the virus. The COVID-19 Vaccine Global Access initiative (COVAX) has been responsible for the distribution of approved vaccines worldwide, especially in developing countries. This program includes the direct purchase of vaccines from producers and donations from the different participating countries. By the end of December 2021, it had delivered more than 850,000,000 doses worldwide [140]. This initiative has been the main form of vaccine delivery in developing countries, especially Africa. However, this number of vaccines corresponds to less than 10% of the total vaccines used worldwide [137]. The remaining percentage has been acquired mainly by higher-income vaccines with the capacity to buy vaccines directly or indirectly from vaccine producers. Therefore, it is essential to increase the arrival of vaccines in these regions of the world to allow the containment of the virus and to prevent the generation of new variants.

### 8.3. Use of Face Masks

Personal protection measures are effective in reducing the contagion and spread of SARS-CoV-2. Multiple observational and interventional studies have shown that masks reduce virus transmission [127]. This benefit is found when the compliance rate is high. These results are consistent with ecological studies that have shown that, in countries where public measures of mandatory use of masks have been developed, there has been a decrease in the rate of viral transmission as compared to those places where this has not been carried out [141]. Therefore, this measure is essential for controlling viral transmission, especially in areas with a high incidence of infection.

However, especially during the first wave of the pandemic, the availability of appropriate masks (N95, surgical, or similar) decreased significantly, with a significant increase in costs. In consequence, the implementation of other measures such as “source control” (use of masks preferentially/exclusively by high-risk people) have been suggested [127]. Observational studies suggest that this measure is effective in protecting against infection. Although its benefits are less clear as compared to the massive use of masks, this strategy could be effective for infection control in countries with low resources.

### 8.4. Identification of Infected Patients and Containment Strategies

The early detection of patients infected by SARS-CoV-2 is a crucial part of the management of the pandemic, both to screen patients at high risk of severe COVID-19, to prevent deaths and to contain the disease spread. Studies that have evaluated the use of population-based measures to contain the spread of the virus indicate that they effectively reduce the spread of the virus, reducing the burden on public health systems [142,143]. This suggest that control measures could effectively prevent the spread of the virus and the formation of new VOIs/VOCs.

Given the severe social and economic consequences of mass quarantines, it is reasonable to limit these containment measures to those groups of people infected during the period of virus transmission. The recommended isolation time for an infected patient varies between 10–14 days depending on the criteria used in different countries [144,145]. Studies have shown that isolation and quarantine reduce the incidence and mortality associated with COVID-19 at the population level [146]. To achieve this, effective and rapid detection of infected patients is needed. The most accurate method is the polymerase chain reaction (PCR). However, this method is scarcely available in many developing countries, especially sub-Saharan African countries.

Within the Access to COVID-19 Tools (ACT) Acceleration [147], a global campaign that includes the WHO, countries, and non-governmental organizations, for the development, production, and equitable access of tests, treatments, and vaccines for COVID-19; UNICEF has developed a program to provide access to a low-cost, high-precision PCR test (less than USD 3.00 per measurement) in low-resource countries [148]. This is allowing the arrival of new virus detection methods to run timely containment programs. These measures seek to stop the spread of the virus, especially of the new variants.

## 9. Conclusions

The COVID-19 pandemic has had a significant impact globally. The magnitude and duration of the pandemic have been extended in part by the development of new variants of SARS-CoV-2 such as lineage B.1.617.2 (Delta variant) with a higher severity of the disease and lower efficacy of vaccines, and the recently described lineage B.1.1.529/BA.1 (Omicron variant) with an increased transmissibility and immune escape as compared to lineage B.1.617.2. The emergence of these new variants has multiple factors involved, including characteristics of the virus such as its mutation rate and changes in the RBD region of the spike protein with the host cell’s receptor, increasing the virus’s infectious capacity. However, there are other additional variables such as geographic and environmental conditions and the impact of social determinants of health, including inequity in public health, poor access to vaccines, and appropriate protection measures. All these conditions influence the existence of regions with higher risk for the emergence of SARS-CoV-2 variants, of potential global risk.

Given this, the WHO and other entities have proposed strategies to prevent the emergence and spread of these new variants, including regional and international policies. These strategies include genomic surveillance of the virus at the local level, access to anti-SARS-CoV-2 vaccines, the availability of face masks, identifying infected patients, and implementing appropriate containment strategies. Implementing these strategies is key to reducing the spread of new variants of SARS-CoV-2, preventing the emergence of new variants of the virus, and finally being able to reach the end of this pandemic.

## Figures and Tables

**Figure 1 life-12-00194-f001:**
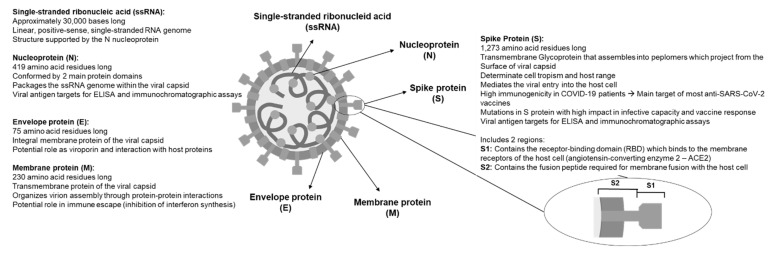
Scheme of the structure of SARS-CoV-2, highlighting its main components and relevant characteristics.

**Figure 2 life-12-00194-f002:**
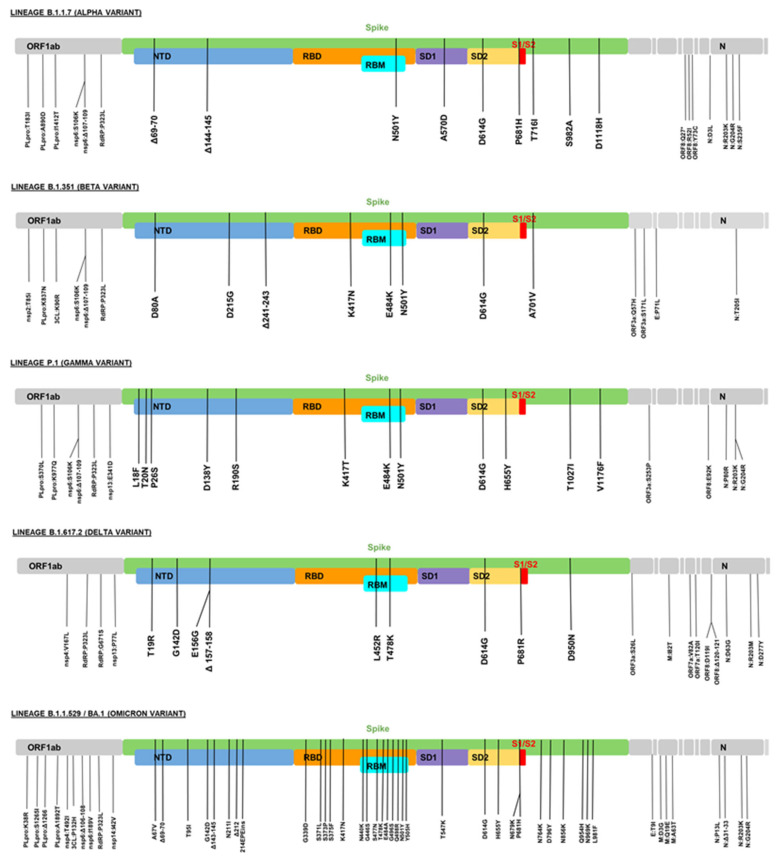
Characteristic mutations (present in more than 75% of the analyzed sequences) of variants of concern of SARS-CoV-2, as of 31 December 2021 (GISAID database).

**Table 1 life-12-00194-t001:** Classification of SARS-CoV-2 variants of the WHO Technical Advisory Group on Virus Evolution.

	Variant of Interest (VOI)	Variant of Concern (VOC)	Variant Under Monitoring (VUM)	Former Monitored Variant
World Health Organization working definition	1. SARS-CoV-2 variant with genetic changes that are predicted or known to affect virus characteristics such as transmisibility, disease severity, immune escape, diagnostic and therapeutic escape. 2. Identified to cause significant community transmission or multiple COVID-19 clusters, in multiple countries with increasing relative prevalence alongside increasing number of cases over time, or other apppernt epidemiological impacts to suggest and emerging risk to global public health.	1. SARS-CoV-2 variant that meets VOI criteria. 2. Associated with one or more of the following characteristics of global public health relevance: (a) Increase in transmisibility or detrimental change in COVID-19 epidemiology. (b) Increase in virulence or change in clinical disease presentation. (c) Decrease in effectiveness of public health and social measures or available diagnostics, vaccines, therapeutics.	SARS-CoV-2 variant with genetic changes suspected to affect virus characteristics with a potential future risk, but unclear evidence of phenotypic or epidemiological impact.	Previous VOCs/VOIs/VUMs that have been reclassified on at least one of the following criteria: (a) The variant is no longer circulating at levels of global public health significance. (b) The variant has been circulating for a long time without any significant epidemiological impact. (c) Scientific evidence demonstrate that the variant is not associated with concerning properties.
Designed SARS-CoV-2 variants as of 31 December 2021 (Pango lineage)	C.37 (Lambda variant) B.1.621 (Mu variant)	B.1.1.7 (Alpha variant) B.1.351 (Beta variant) P.1 (Gamma variant) B.1.617.2 (Delta variant) B.1.1.529/BA.1 (Omicron variant)	AZ.5 C.1.2 B.1.617.1 * B.1.525 * B.1.526 * B.1.630 B.1.640	AV.1 AT.1 P.2 * P.3 * R.1 B.1.466.2 B.1.1.519 C.36.3 B.1.214.2 B.1.427/B.1.429 * B.1.1.523 B.1.619 B.1.620

* Previously classified as VOIs, then reclassified as Variant Under Monitoring or Former Monitored Variant.

**Table 2 life-12-00194-t002:** General characteristics of the SARS-CoV-2 variants of concern (VOC) as of 31 December 2021 by the World Health Organization.

WHO Label	Pango Lineage	GISAID Clade	Nextstrain Clade	Origin of First Samples Detected	Date of First Documented Case	VOC Designation Date	Transmissibility	Clinical Severity	Clinical Response to Anti-SARS-CoV-2 Vaccines
Alpha	B.1.1.7	GRY	20I (V1)	United Kingdom	sept-20	18-12-2020	Higher than previous variants (virological and epidemiological evidence)	Possibly similar incidence of severe COVID-19 and death (epidemiological studies with mixed results)	Vaccines prevent infection and adverse events
Beta	B.1.351	GH/501Y.V2	20H (V2)	South Africa	sept-20	18-12-2020	Higher than previous variants (virological and epidemiological evidence)	Possibly similar incidence of severe COVID-19 and death (epidemiological studies with mixed results)	Vaccines prevent infection and adverse events (possibly less effective in preventing infection)
Gamma	P.1	GR/501Y.V3	20J (V3)	Brazil	nov-20	11-01-2021	Higher than previous variants (virological and epidemiological evidence)	Possibly similar incidence of severe COVID-19 and death (epidemiological studies with mixed results)	Vaccines prevent infection and adverse events (possibly less effective in preventing infection)
Delta	B.1.617.2	G/478K.V1	21A, 21I, 21J	India	oct-20	11-05-2021 *	Much higher than previous variants (virological and epidemiological evidence)	Increased incidence of severe COVID-19 and death (virological and epidemiological evidence)	Vaccines prevent infection and adverse events (slightly lower efficacy in preventing infection)
Omicron	B.1.1.529 BA.1 ^#^	GRA	21K, 21L, 21M	South Africa	nov-21	26-11-2021 **	Extremely higher than previous variants (virological and epidemiological evidence)—higher than lineage B.1.617.2	Preliminary data suggests that it has a lower incidence of severe COVID-19 and death—lower than lineage B.1.617.2	Preliminary data suggest that vaccines prevent infection and adverse events (possible lower efficacy versus other VOCs)

* Initially designated as VOI (04-04-2021); ** Initially designated as VUM (24-11-2021); ^#^ Change of classification proposed by the Pango Network to differentiate from the BA.2 sublineage (without del69-70).

**Table 3 life-12-00194-t003:** World Health Organization recommendations for managing of a new variant of interest (VOI) or a variant of concern (VOC).

	Variant of Interest (VOI)	Variant of Concern (VOC)
Primary actions by World Health Organization (WHO) for a potential VOI/VOC	Comparative assessment of variant characteristics and public health risks by WHO.	Comparative assessment of variant characteristics and public health risks by WHO and the Technical advisory Group on Virus Evolution.
	If determined necessary, coordinated laboratory investigations with Member States and partners.	If determined necessary, coordinate additional laboratory investigations with Member States and partners.
	Review global epidemiology of VOI.	Communicate new designations and findings with Member States and public through established mechanisms.
	Monitor and track global spread of VOI.	Evaluate WHO guidance through established WHO mechanisms and update, if necessary.
Primary actions by a Member State if a new potential VOI/VOC is identified	Inform WHO through established WHO Country or Regional Office reporting channels with supporting information about VOI-associated cases (person, place, time, clinical and other relevant characteristics).	Submit complete genome sequences and associated metadata to a publicly available database, such as GISAID.
	Submit complete genome sequences and associated metadata to a publicly available database, such as GISAID.	Report initial cases/clusters associated with VOC infection to WHO through the International Health Regulations (IHR) mechanism.
	Perform field investigations to improve understanding of the potential impacts of the VOI on COVID-19 epidemiology, severity, effectiveness of public health and social measures, or other relevant characteristics.	Where capacity exists and in coordination with the international community, perform field investigations and laboratory assessments to improve understanding of the potential impacts of the VOC on COVID-19 epidemiology, severity, effectiveness of public health and social measures, diagnostic methods, immune responses, antibody neutralization, or other relevant characteristics.
	Perform laboratory assessments according to capacity or contact WHO for support to conduct laboratory assessments on the impact of the VOI on relevant topics.	

## Data Availability

Not applicable.

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
