# Peer review of "Emergence of SARS-CoV-2 Variants in the World: How Could This Happen?"

_life, 2022, doi:10.3390/life12020194_

Round 1
Reviewer 1 Report
line 63 - check the given example
line 64 to 83 and line 95 to 109 - is repeating the information from table 1 and possibly could be removed
line 146 check citation
line 172 - please check for more recent literature on the Mutation rate for SARS-CoV-2, i found numbers more like 8x10^-4, which is normal or little lower than for other RNA viruses
line 187 - please screen for better fitting literature on recombination events as mentioned
Table 3 and figure 2 are redundant, may choose one (correct L452R Mutation for Delta in Fig.2)
line 238 - if you wanne follow up virus evolution and diversification you should also mention the global spread of D614G variant before the spread of Alpha (https://doi.org/10.1016/j.cell.2020.06.043)
line 259 - may to mention , that there is actually no beta circulating anymore
line 631 - omicron is far more immune / vaccination escaping than delta
line 632 - not unusual high mutation rate (see also proof reading polymerase)
please check once more that you have for every claim appropriate citation(s)
Author Response
Answer to Reviewer in .doc file

Reviewer 2 Report
An excellent and very comprehensive review.
in Point 7: Is it possible a Country's effort in controlling the disease / pandemic also contribute to the emergent of VOC?
Point 8: Strategies in control: A collaborative effort of global vaccination is surely useful, however any suggestions in how to implement it?
Author Response
Answer to Reviewer in .doc file
